# BDS/GPS/Galileo Precise Point Positioning Performance Analysis of Android Smartphones Based on Real-Time Stream Data

**Mengyuan Li [1], Guanwen Huang [1,2,*], Le Wang [1] and Wei Xie [1]**

1   College of Geology Engineering and Geomatics, Chang'an University, Xi'an 710054, China; limy@chd.edu.cn (M.L.); wangle18@chd.edu.cn (L.W.); chdxiewei@chd.edu.cn (W.X.)

2   Key Laboratory of Ecological Geology and Disaster Prevention, Ministry of Natural Resources, Xi'an 710054, China

\*   Correspondence: guanwen@chd.edu.cn; Tel.: +86-13636801167

**Abstract:** Smartphones with the Android operating system can acquire Global Navigation Satellite System (GNSS) raw pseudorange and carrier phase observations, which can provide a new way for the general public to obtain precise position information. However, only postprocessing precise orbit and clock offset products in some older smart devices are applied in current studies. The performances of precise point positioning (PPP) with the smartphone using real-time products and newly smartphones are still unrevealed, which is more valuable for real-time applications. This study investigates the observation data quality and multi-GNSS real-time PPP performance using recent smartphones. Firstly, the observed carrier-to-noise density ratio (C/N0), number of satellites and position dilution of precision (PDOP) of GNSS observations are evaluated. The results demonstrate that the C/N0 received by Huawei Mate40 is better than that of the Huawei P40 for GPS, BDS, QZSS and Galileo systems, while the GLONASS is poorer, and the PDOP of the Huawei P40 is slightly better than that of Mate40. Additionally, a comprehensive analysis of real-time precise orbit and clock offset products performance is conducted. The experiment result expresses that the orbit and clock offset performance of GPS and Galileo is better than that of BDS-3 and GLONASS, and BDS-2 is the worst. Finally, single- and dual-frequency multi-GNSS combined PPP experiments using observations received from smartphones and real-time products are conducted; the results indicate that the real-time static PPP using a smartphone can achieve decimeter-level positioning accuracy, and kinematic PPP can achieve meter-level positioning accuracy after convergence.

**Keywords:** real-time streaming; smartphone; PPP; single- and dual-frequency; BDS/GPS/GALILEO

## 1. Introduction

Nowadays, real-time positioning services play an important role in several areas, such as public navigation [1]. The real-time-positioning information of users relies on the GNSS real-time satellite orbit, clock offset products on the server and the GNSS measurements of users [2]. The real-time products from the International GNSS Service (IGS) analysis centers are transmitted through the Networked Transport of RTCM via Internet Protocol (NTRIP). In addition, the real-time products from many IGS analysis centers are uploaded to Caster by the server, and users can access Caster to obtain real-time products through the Client [3]. The performance of GNSS real-time orbit and clock offset products has been greatly improved in recent years, and in terms of receiving GNSS measurements, users can use various types of receivers to receive real-time observations. Additionally, smartphones have become indispensable tools in daily life due to their high popularity. A low-cost GNSS chip was embedded in several smartphones, which can receive GNSS measurements [4]. In 2016, the operating system of Android N was released by Google; since then, an interface has been provided by Android smart devices, and the raw GNSS

observations could be exported [5]. Therefore, real-time positioning services based on the smartphone are possible.

The study of GNSS smartphones can be divided into observation data quality evaluation and positioning performance analysis. Several studies have already researched the observation data quality assessment of smartphones. Zhang et al. evaluated the carrier-to-noise density ratio (C/N0) and pseudorange noise at GPS L1 frequency received by the Google/HTC Nexus 9 smartphone; the C/N0 was 10 dB-Hz lower than the geodetic-quality receiver, and the single-difference pseudorange residuals were between −20 m and 20 m [6]. Liu et al. compared the C/N0 and single-difference pseudorange residuals between short baseline stations and three smart devices, including the Samsung Samsung Galaxy S8, Huawei P10 and Google Nexus 9 tablet. The single-difference pseudorange was 3.09, 6–7 and 12–13 m for BDS, GPS and GLONASS, respectively [7]. Paziewski et al. evaluated the C/N0 and pseudorange noise of the Huawei P20 smartphone, and the C/N0 was 9.4 dB-Hz lower than that of the Javad Alpha receiver, and the pseudorange noise was beyond 10 times higher than the receiver [8]. Zhu et al. found that the C/N0 were larger than 10 dB-Hz, and the percentages of the cycle slip rate for GPS, GLONASS and BDS are approximately 1.85%, 4.04% and 0.18%, respectively [9].

In terms of positioning using a smartphone, Wang et al. used a Samsung Galaxy A7 smartphone to conduct BDS/GPS static and kinematic real-time kinematic (RTK) positioning; the static RTK positioning accuracy was better than 0.15 m and 0.25 m in horizontal and vertical directions, respectively, while kinematic RTK was better than 0.30 m and 0.45 m in horizontal and vertical directions, respectively [10]. Robustelli et al. employed GPS L1 observations from a Xiaomi 8 mobile phone to perform PPP; the accuracies were 0.47 m, 0.91 m and 0.5 m in the east (E), north (N) and up (U) directions, respectively [11]. Guo et al. showed that the positioning accuracy of single-frequency PPP could reach 0.5–0.6 m horizontally and 1.0–2.0 m vertically for Xiaomi 8 and Huawei Mate20/30 mobile phones [12]. Wanninger et al. analyzed the GPS L1 signal from a Huawei P30 smartphone for differential code solution and were able to achieve positioning accuracies of 0.006, 0.005 and 0.015 m in the N, E and U directions, respectively, after one hour of continuous observation [13]. Zhu et al. established a smartphone multi-GNSS PPP model; the static PPP accuracy was 0.188, 0.165 and 0.761 m in the E, N and U directions, respectively, while the kinematic PPP was 0.928, 0.624 and 2.167 m [9]. The best horizontal and vertical positioning errors achievable for the Xiaomi 8 smartphone with GPS, BDS and Galileo systems were, on average, 1.65 and 8.1 m in Chen et al. [14]. Berkay et al., based on the observations received from three smartphones (Xiaomi 8, Google Pixel 4 and Pixel 4 XL), the multi-GNSS PPP was conducted using ultra-rapid and IGS Real-Time Service (IGS-RTS) products, respectively. The PPP accuracy using IGS-RTS can be improved by about 0.11%, 2.8% and 4.4% compared to ultra-rapid products for Xiaomi 5, Pixel 4 and Pixel 11XL smartphones, respectively [15]. Li et al. proposed a real-time PPP method for smart devices for in-vehicle motion localization. The RMS error of the positioning results obtained based on real-time PPP was about 1.0–1.5 m in the horizontal direction and 1–2 m in the vertical direction when the smartphone was placed in the vehicle [16].

It can be clearly seen that most of the above studies are based on post-processing precise orbit and clock offset for post-processing PPP and RTK. Fewer studies have been conducted on the comparative evaluation of the PPP performance based on real-time streaming data of recent mobile phones. Because there is a time delay for post-processing precise orbit and clock offset products, which cannot meet the real-time application. In addition, previous studies have used older smart devices, such as Xiaomi 8, while little research has reported on the real-time PPP performance based on single- and dual-frequency and multi-GNSS of relatively new smartphones, such as the Huawei Mate40 and P40; the positioning performance of these newer smartphones is still unrevealed [17–20]. Therefore, the data quality and real-time positioning performance using Huawei Mate40 and P40 should be further investigated.

This contribution investigates the multi-GNSS observation data quality evaluation and PPP performance based on the real-time data stream of the smartphone. Based on the observation data from Huawei Mate40 and P40 and geodetic receiver and the real-time precise orbit and clock offset products from GFZ on 16 November 2022, the observation data quality of the Huawei Mate40 and P40 smartphones and the real-time satellite orbit and clock offset accuracy were assessed, and the real-time PPP performance using smartphones was evaluated. After this introduction, the real-time smartphone PPP model was introduced. Then the data quality of the smartphone was evaluated in terms of C/N0, the number of satellites and PDOP. Furthermore, we analyzed the performance of real-time orbit and clock offset. Thereafter, we investigated the real-time static and kinematic PPP performance of smartphones. Finally, the conclusions are given.

## 2. Methodology of Real-Time PPP for the Smartphone

Real-time PPP of the smartphone refers to the use of real-time orbit and clock offset products for high-precision positioning based on pseudorange and carrier-phase observations measured by the smartphone and combined with a tight error correction model [21,22]. The real-time multi-GNSS undifferenced and uncombined PPP model is shown in Equation (1).

$$\begin{cases} P_{r,j}^{s,Q} = \rho_r^{s,Q} + c\left(dt_r - dt^{s,Q}\right) + T_r^{s,Q} + I_r^{s,Q} + c\left(d_{r,j}^{s,Q} - d_j^{s,Q}\right) + \varepsilon_{P_{r,j}}^{s,Q} \\ L_{r,j}^{s,Q} = \rho_r^{s,Q} + c\left(dt_r - dt^{s,Q}\right) + T_r^{s,Q} - I_r^{s,Q} + \lambda_j^{s,Q}\left(N_{r,j}^{s,Q} + b_{r,j}^{s,Q} - b_j^{s,Q}\right) + \varepsilon_{L_{r,j}}^{s,Q} \end{cases} \tag{1}$$

where $s$ and $Q$ denote the satellite pseudo-random noise (PRN) and system, in which G, C, R and E represent GPS, BDS, GLONASS and Galileo systems, respectively. $r$, $j$ are the receiver and observation frequencies, respectively. $P_{r,j}^{s,Q}$ and $L_{r,j}^{s,Q}$ express pseudorange and carrier observations, respectively. $\rho_r^{s,Q}$ shows the geometric distance between the station and the satellite. $c$ is the speed of light propagation in a vacuum; $dt_r$ and $dt^{s,Q}$ signify the receiver and the satellite clock offset, respectively. $T_r^{s,Q}$ and $I_r^{s,Q}$ denote tropospheric and ionospheric delay, respectively. $d_{r,j}^{s,Q}$ and $d_j^{s,Q}$ are the pseudorange hardware delay at the receiver and satellite side, respectively. $\lambda_j^{s,Q}$ represents the carrier phase wavelength. $N_{r,j}^{s,Q}$ denotes the carrier phase ambiguity. $b_{r,j}^{s,Q}$ and $b_j^{s,Q}$ express the phase hardware delay at the receiver and the satellite, respectively. $\varepsilon_{P_{r,j}}^{s,Q}$ and $\varepsilon_{L_{r,j}}^{s,Q}$ are the sum of observation noise and multipath and other unmodelled errors for pseudorange observations and carrier phase observations, respectively.

In this contribution, for single-frequency PPP, the undifferenced and uncombined model of the smartphone is applied to estimate the station coordinates, and the ionospheric delay is also estimated [23,24], while for the dual-frequency PPP of the smartphone, the ionosphere-free combination PPP model is employed [25]. The parameters that should be estimated include station coordinates, receiver clock offset, tropospheric wet delay and ambiguities [26–29]. Smartphone real-time PPP calculation includes data preprocessing module, error correction module and parameter estimation module. The specific solution process is shown in Figure 1. The detailed processing strategies for smartphone real-time PPP are listed in Table 1.

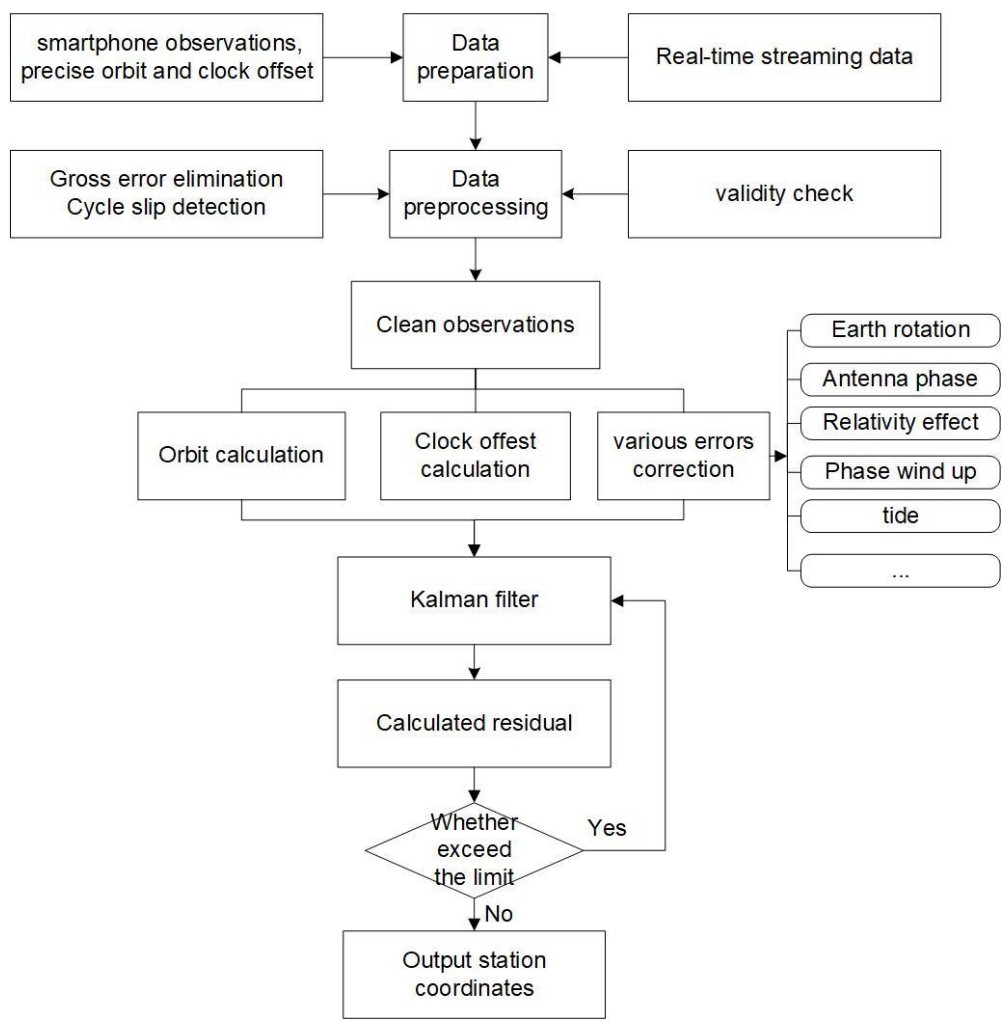

**Figure 1.** Flow chart of real-time PPP solution with smartphones.

**Table 1.** Real-time PPP of the smartphone processing strategy.

| | Items | Strategy/Model |
|---|---|---|
| Observations | Processing models | Single frequency:<br>GPS: L1; BDS: B1I; Galileo: E5a<br>Dual frequency: GPS: L1/L5 |
| | Sampling interval | 1s |
| | Cut-off angle | 15° |
| Correction | Tropospheric delay | Saastamoinen model and GMF function |
| | Satellite hardware delay | CODE products |
| | Satellite PCO and PCV | igs14.atx |
| | Satellite orbit/clock deviation | GFZ real-time product |
| | Phase windup | Model correction |
| | Relativistic effect | Model correction |
| | Earth rotation | Model correction |
| Parameter estimation | Station coordinates | Static: constant; Kinematic: white noise |
| | Tropospheric delay | Random walk |
| | Receiver clock offset | White noise |
| | ambiguity | Constant (float solution) |
| | Inter-system bias | Random walk |

### 3. Data Collection and Observation Quality Assessment

The experiment was conducted on the roof of the main teaching building at Chang'an University, located in Xi'an City, Shaanxi Province. The experimental environment had an uncovered view and favorable observation conditions, which are shown in Figure 2. The experiment equipment was conducted using Huawei Mate40, Huawei P40 and Novatel receiver named DH04, and the smartphone was set to upright. The received signal frequencies of the two smartphones are shown in Table 2. The Huawei Mate40 and P40 can receive GPS L1 + L5 dual frequency, GLONASS L1C, BDS B1I, Galileo E5a and QZSS L1 + L5 dual-frequency signals. Furthermore, in real-time processing, we received the real-time stream of multi-GNSS broadcast ephemeris and satellite orbit and clock offset products from the NTRIP Caster. The experiment was conducted on 16 November 2022 at a sampling rate of 1 s and a total duration of approximately 2 h from 10:00 to 12:00 UTC, with the use of Geo++RINEX Logger. The mean coordinates of the last 500 s were used as the reference position.

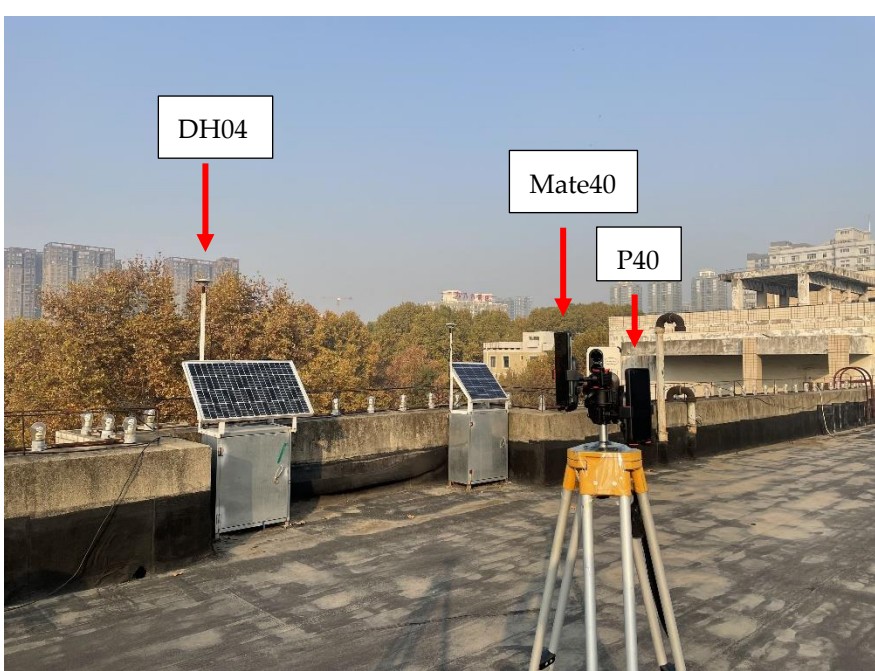

**Figure 2.** Experimental environment.

**Table 2.** Smartphone configurations.

|  | **Huawei Mate40** | **Huawei P40** |
| --- | --- | --- |
| CPU | Kylin 9000E | Kylin 9905G |
| frequency | GPS (L1 + L5)/GLONASS (L1C)/BDS (B1I)/Galileo (E1 + E5a) | |
| Release Time | October 2020 | March 2020 |
| Appearance | | |

### 3.1. Carrier-to-Noise Density Ratio

The C/N0 is influenced by many factors, such as the antenna gain parameters, the state of the receiver correlator and the multipath effect, which can directly reflect the GNSS observations data quality [30]. The larger the C/N0 means the stronger the signal; vice versa, the higher the probability of anomalous observations. In this assessment, the C/N0 is classified for the elevation of every 5 degrees, and the average value of C/N0 within 5 degrees of elevation is calculated. Figures 3 and 4 display the average value versus the elevation angle of multi-GNSS C/N0 from Huawei Mate40 and P40 at GPS S1C and S5Q, BDS S2I, GLONASS S1C, QZSS S1C and S5Q and Galileo S5Q, respectively, in which different colors represent the C/N0 received by the different satellites. As shown in the figure, the C/N0 of the smartphone shows periodic fluctuations, the positive correlation between C/N0 and elevation is not found, and the C/N0 of some observations is low at a higher elevation [2]. In terms of systems, the C/N0 value of GPS and QZSS is the best, with an average of about 35 dB-Hz; BDS and GLONASS are poorer, as their mean C/N0 is about 33 dB-Hz, and Galileo system presents the worst performance, its mean C/N0 around 30 dB-Hz. Moreover, the degree of dispersion for smartphone C/N0 is larger than for geodetic receivers, which is related to the omnidirectional, passive, linearly polarized and uneven gain built-in GNSS antenna of the smartphone. Compared to the Huawei P40, the C/N0 value received by the Huawei Mate40 is better for GPS, BDS, QZSS and Galileo observations, while for the GLONASS S1C frequency, the Huawei P40 outperforms the Mate40 by approximately 5 dB-Hz.

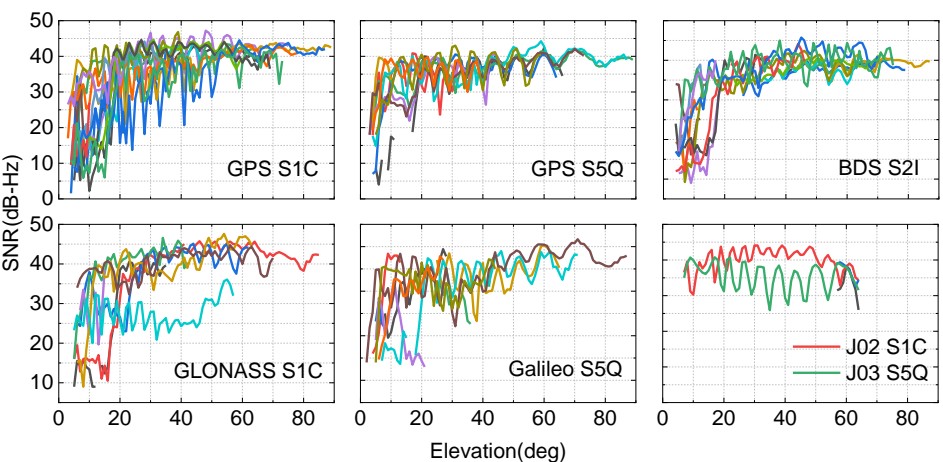

**Figure 3.** Average C/N0 with elevation for the Huawei Mate40.

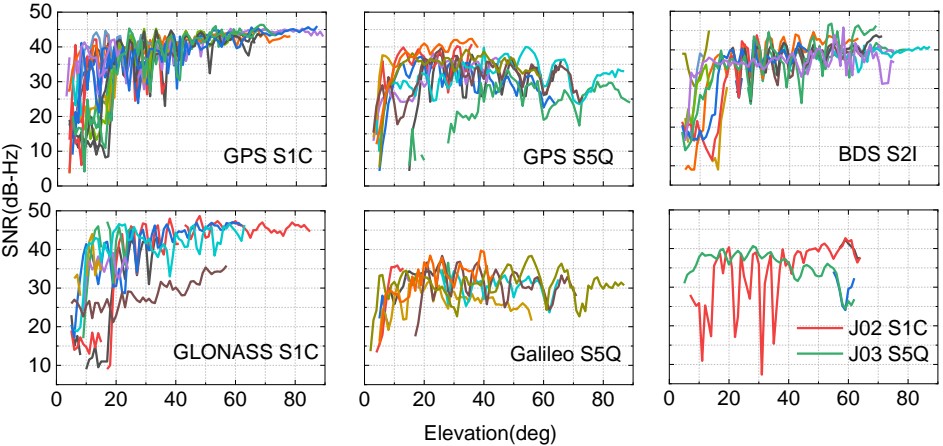

**Figure 4.** Average C/N0 with elevation for the Huawei P40.

### 3.2. The Number of Satellites and PDOP

There is a relationship between tracked satellites and the observation data quality. When the number of tracked satellites is larger, it illustrates a better observation quality [31]. Figure 5 shows the number of satellites that can be observed by the Huawei Mate40, P40 and Novatel receiver, as shown in the figure, and the total number of satellites that can be observed is nearly 25. Overall, the number of observed BDS satellites is larger than that of GPS, while the number of satellites observable by the Galileo system is relatively small. The reason can be attributed to the BDS-2 and BDS-3 ability to provide services, and many satellites can be used to conduct Positioning, Navigation and Time (PNT) experiments. Our experiment was conducted in China, so the number of observed BDS satellites is high. The average number of observed BDS satellites for smartphones is around 15, while it is 8 for GPS satellites. Moreover, the fluctuations in the number of observed satellites for smartphones are larger.

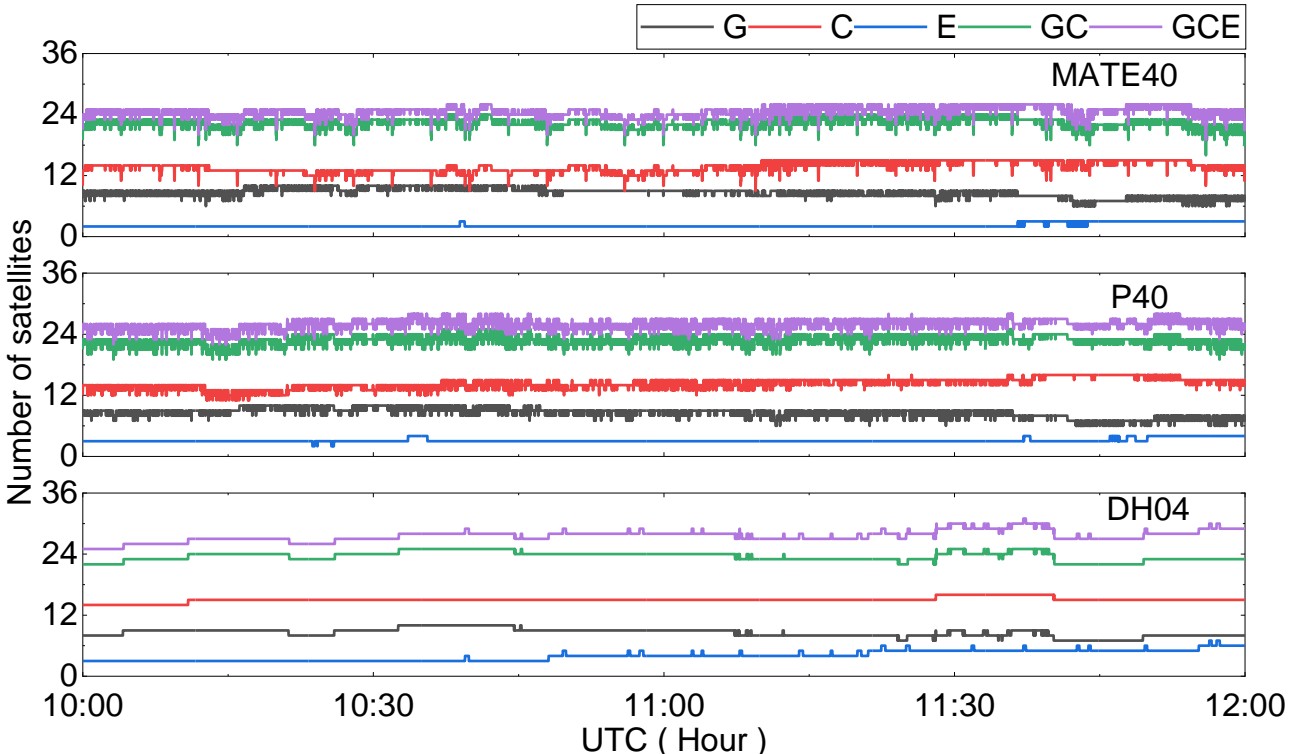

**Figure 5.** Number of satellites for Huawei Mate40 and P40.

The PDOP can reflect the strength of the satellite geometric distribution between the ground-tracked station and the satellite; moreover, the feasibility and accuracy of the positioning can also be reflected, and it is an important indicator to determine the reliability of the GNSS measurements. The smaller the PDOP, the stronger the satellite geometry configuration and the smaller the positioning error, i.e., the higher the accuracy, and vice versa. Figure 6 shows the PDOP of the Huawei Mate40, P40 and Novatel receiver. As can be seen from the figure, the PDOP of the Huawei P40 is slightly better than that of the Huawei Mate40, indicating the better satellite geometry configuration for Huawei P40. The BDS PDOP for smartphones is superior to GPS, and the mean PDOP for both GPS/BDS and GPS/BDS/Galileo is less than 1.8.

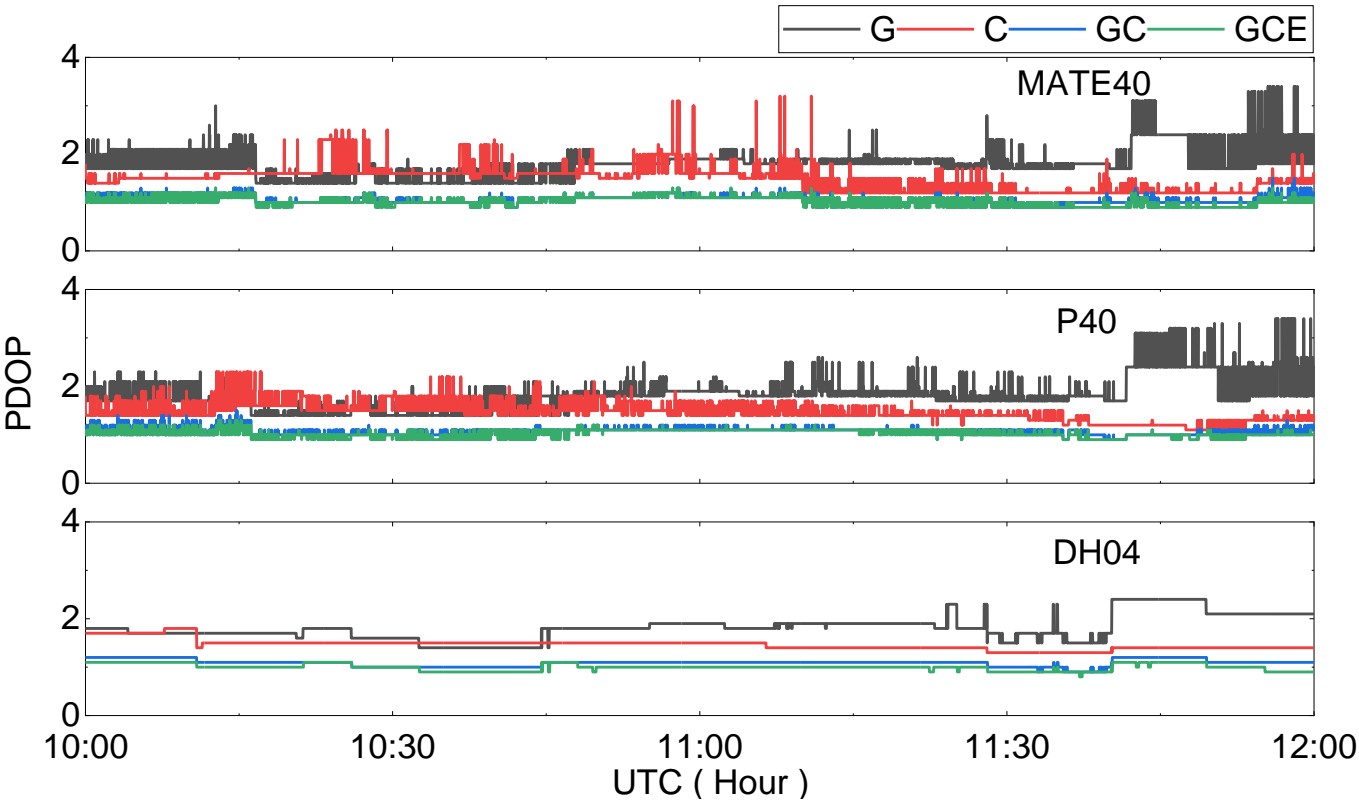

**Figure 6.** PDOP for Huawei Mate40 and P40.

## 4. Real-Time Orbit and Clock Offset Performance

### 4.1. Real-Time Orbit Performance

The satellite orbit product performance is evaluated between real-time and final products in terms of 'along', 'cross' and 'radial' directions; the evaluation methodology is referenced in [32]. The real-time GNSS satellite orbit accuracies in along, cross and radial directions on the day of year (DOY) 319 in 2022 are depicted in Figure 7. The mean accuracies of real-time GNSS satellite orbit in three directions and the three-dimensional (3D) Root Mean Square (RMS) are counted in Table 3.

**Table 3.** The average accuracy of the real-time orbit (unit: m).

| System | Along | Cross | Radial | 3D RMS |
|---|---|---|---|---|
| GPS | 0.064 | 0.029 | 0.025 | 0.075 |
| BDS-2 MEO | 0.127 | 0.055 | 0.030 | 0.142 |
| BDS-2 IGSO | 0.770 | 0.173 | 0.409 | 0.889 |
| BDS-3 MEO | 0.123 | 0.047 | 0.051 | 0.141 |
| BDS-3 IGSO | 1.513 | 0.329 | 1.091 | 1.895 |
| GLONASS | 0.132 | 0.036 | 0.042 | 0.143 |
| Galileo | 0.067 | 0.031 | 0.034 | 0.081 |

As can be seen from Figure 7, for GPS, the satellite orbit accuracy difference for different satellites is small, and the orbit accuracy for most satellites is better than 0.05 m for along, cross and radial directions, while the orbit accuracy of some satellites is larger than 0.05 m for radial direction, such as the G14 satellite. The orbit accuracy of the BDS-2 satellites is poor, particularly in some of the IGSO satellites, which is mainly because of the worse geometry configuration of IGSO satellites [33]. Moreover, the solar radiation pressure model is mainly for MEO satellites, which does not fully satisfied the precise orbit determination of IGSO satellites. Compared with BDS-2, the real-time orbit accuracy of BDS-3 has been significantly improved due to their better geometry configuration and the

improved onboard atomic clock and signal designation. For the BDS-3 IGSO satellites, i.e., C38, C39 and C40, their orbit accuracy is significantly worse than that of BDS-3 MEO satellites. The performance of the real-time orbit among the Galileo satellites is small, and the accuracy in three directions is better than 0.1 m, except for individual satellites. The difference in the real-time orbit accuracy for GLONASS satellites is smaller in along and cross directions, while it is larger in radial directions.

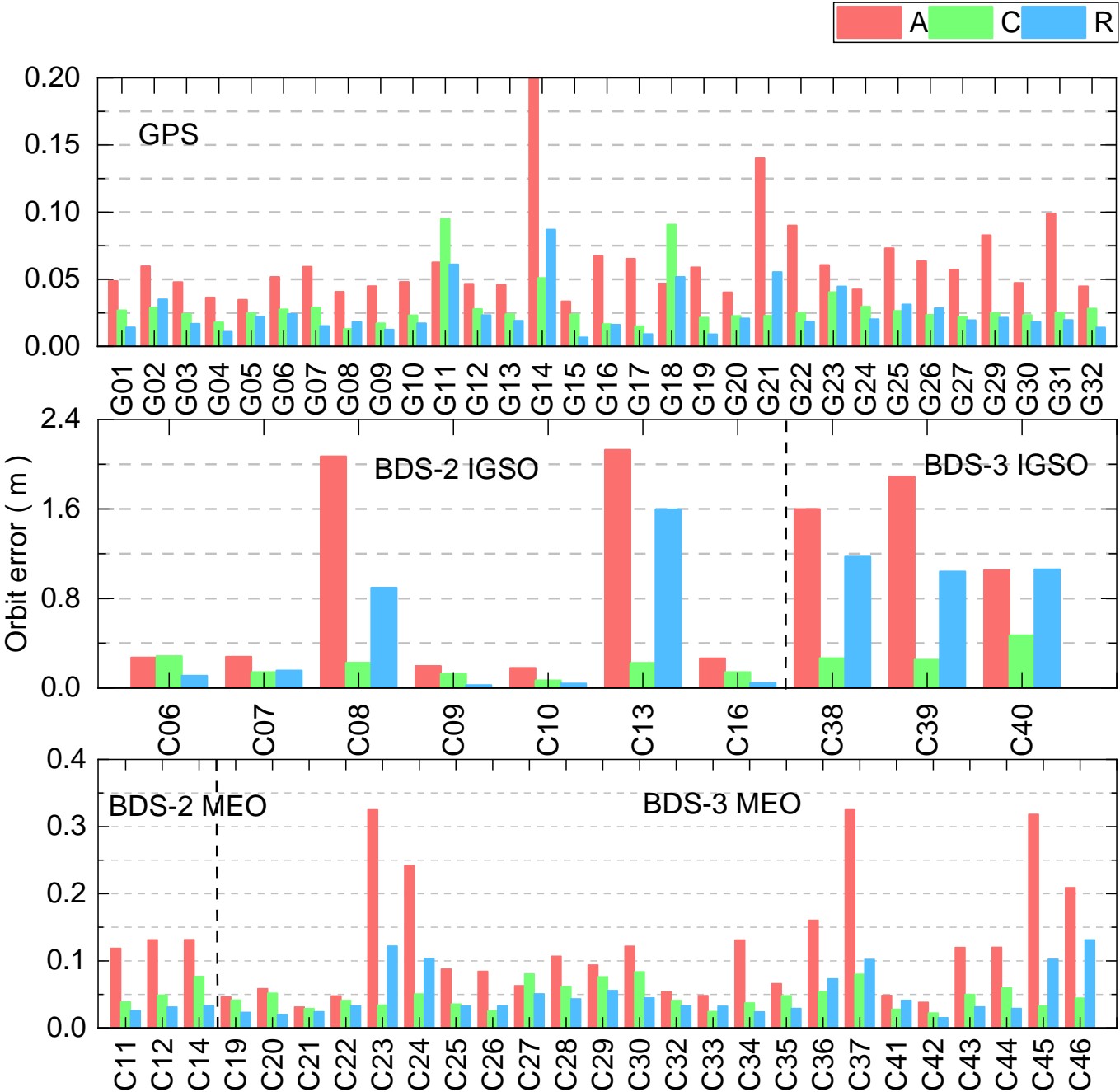

**Figure 7.** *Cont.*

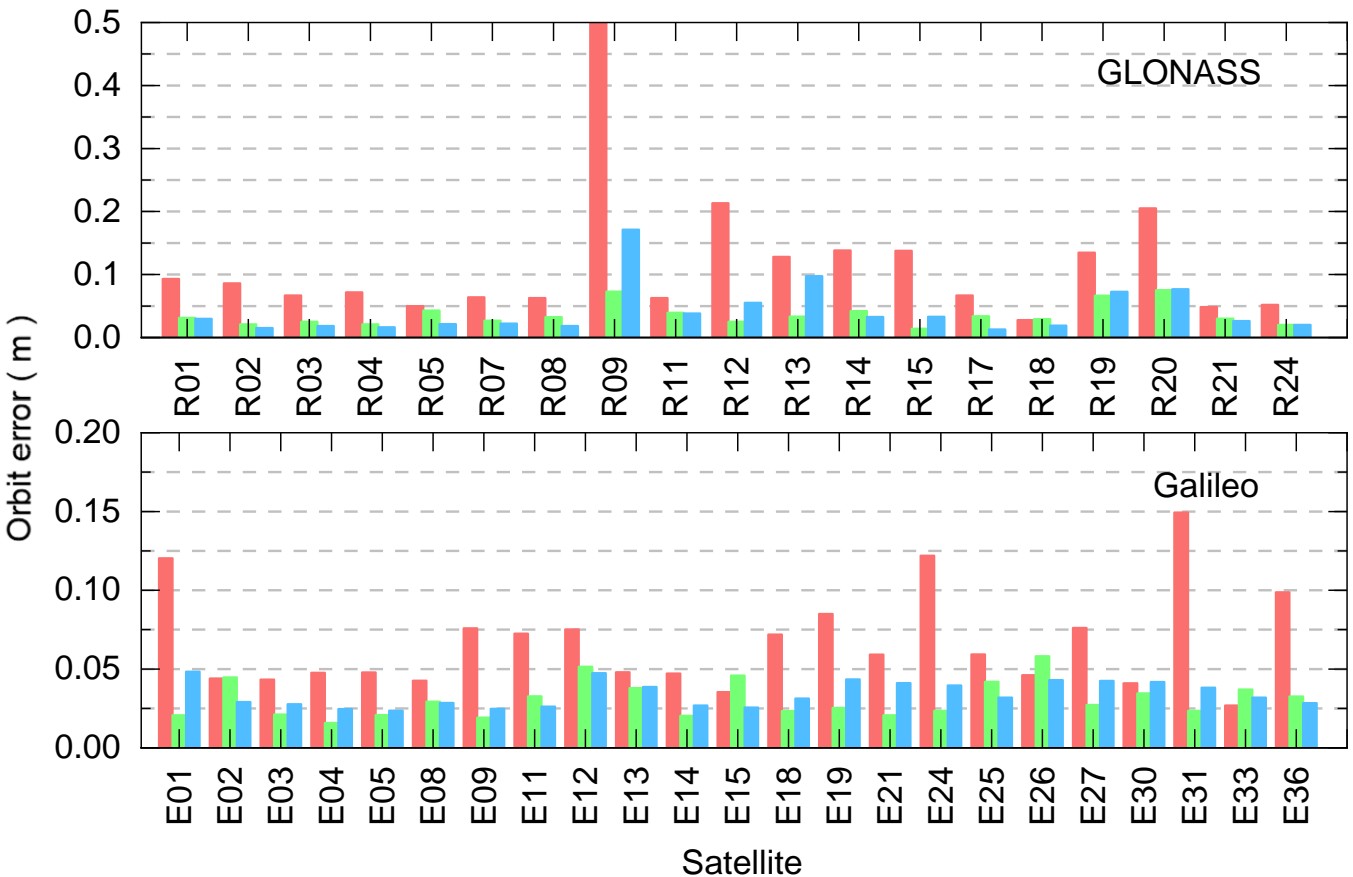

**Figure 7.** Real-time orbit accuracy for GNSS.

It becomes clear that from the statistics given in Table 3, the real-time orbit performances of GPS and Galileo are superior, the BDS MEO and GLONASS satellites are the next best, and the BDS IGSO satellites are the worst.

### 4.2. Real-Time Clock Offset Performance

The post-processing products from the CODE analysis center were chosen as a reference to evaluate the real-time satellite clock offset accuracy, and the assessment method was referred to in [34]. The real-time clock offset accuracy for each satellite is displayed in Figure 8. As can be seen from the figure, the clock offset accuracy of most GPS and Galileo satellites is better than 0.15 and 0.25 ns, respectively. For BDS, impacted by satellite orbit geometry, the clock offset accuracy of BDS-2 IGSO satellites is worse than the MEO satellites. Compared to BDS-2, the clock offset accuracy of the BDS-3 satellite is significantly improved. On the one hand, there are more MEO satellites for BDS-3, and more satellites can be tracked for stations located outside the Asia-Pacific region, which is beneficial for clock offset estimation; on the other hand, the improved rubidium atomic clocks and passive hydrogen atomic clocks were equipped on BDS-3 satellites, and the frequency stability of the atomic clocks was considerably improved [35]. For the GLONASS satellites, the clock offset performance is better than 0.6 ns, except for R09.

The average clock offset accuracy for all satellites is shown in Table 4; the clock offset accuracy is 0.158, 0.163, 0.553, 0.280 and 0.317 ns for GPS, Galileo, BDS-2, BDS-3 and GLONASS satellites, respectively.

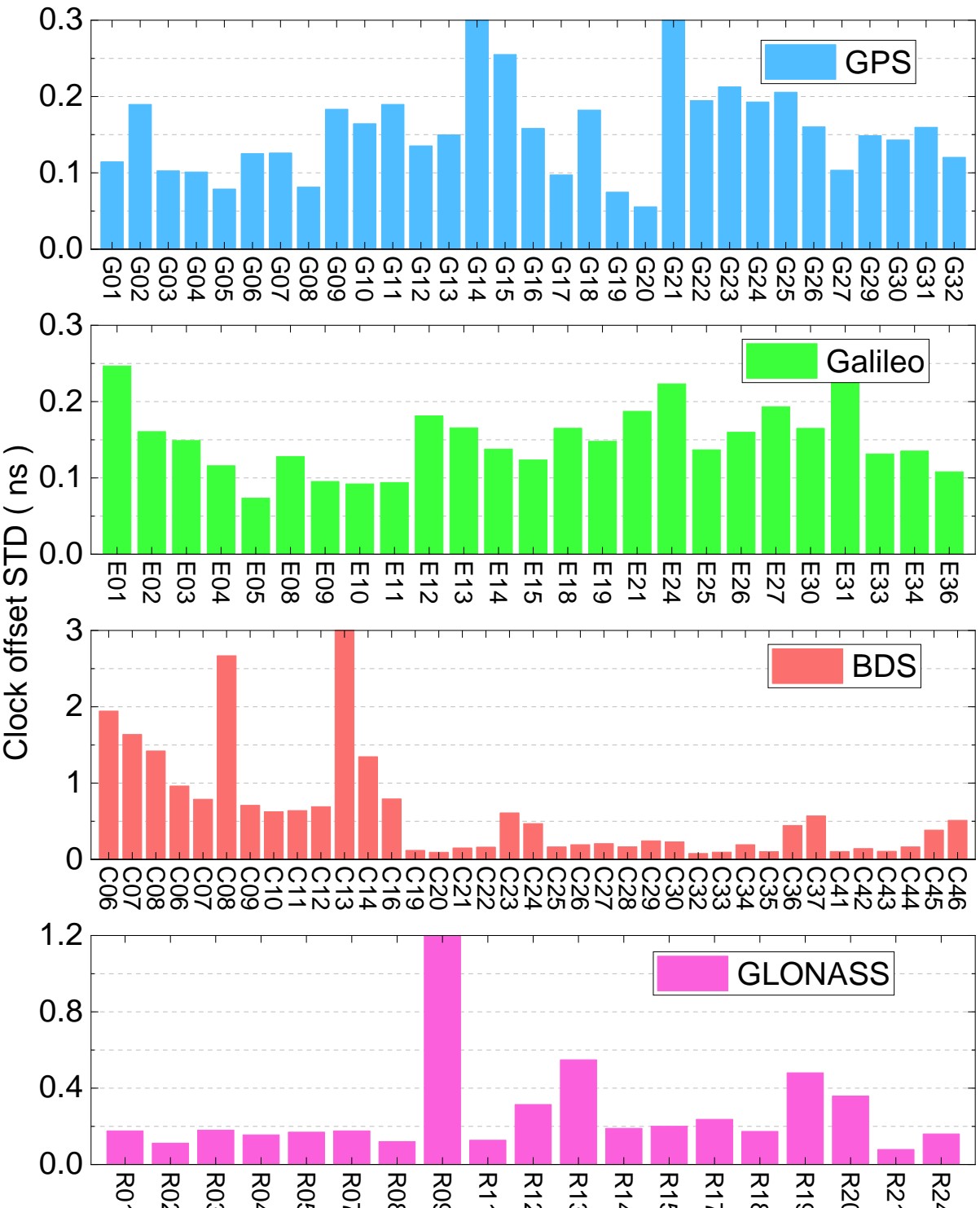

**Figure 8.** Real-time clock offset off accuracy for GNSS.

**Table 4.** The average accuracy of the real-time clock offset (unit: ns).

|  | GPS | Galileo | BDS-2 | BDS-3 | GLONASS |
|---|---|---|---|---|---|
| clock offset STD | 0.158 | 0.163 | 0.553 | 0.280 | 0.317 |

## 5. Real-Time PPP Performance of Smartphone

In this experiment, because of the lack of carrier observations for GLONASS satellites in the received observation data, only the real-time PPP performance of the smartphone of the GPS, BDS, GPS/BDS combination, GPS/BDS/Galileo combination and GPS L1/L5 combination were evaluated.

### 5.1. Static PPP Performance

The observations received by the Huawei Mate40, P40 and Novatel receiver on DOY 319 were for the period of 10:00–12:00 UTC. After obtaining the coordinates, they were converted to E, N and U components to evaluate the smartphone positioning performance. Thus, the convergence time and positioning accuracy were used to evaluate the smartphone PPP performance. The convergence is defined as the RMS coordinates being less than 0.3 m for the current and next 20 consecutive epochs. The static positioning experiments were conducted for GPS, BDS, GPS/BDS combination, GPS/BDS/Galileo combination and GPS L1/L5 combination, respectively. The performance differences between single-system and multi-system combined positioning of smartphones and geodetic receiver were compared.

The positioning error of static real-time PPP for five combinations using Huawei Mate40, P40 and Novatel receivers in the E, N and U components are presented in Figure 9 (in this figure, HM40, HP40 and DH04 represent Huawei Mate40, Huawei P40 and Novatel receiver, respectively). As can be seen from Figure 9, the positioning errors differences between the smartphone and geodetic receiver are slightly large; the positioning errors for the geodetic receiver are more stable than that of the smartphone because the measurement noise of the geodetic receiver is less than that of the smartphone. For the smartphone real-time static PPP, centimeter-level positioning accuracy can be achieved after convergence. For the Huawei Mate40, the fluctuation of positioning accuracy for the GPS-only and BDS-only solutions is larger, and the convergence times are long. However, compared to GPS-only and BDS-only solutions, the convergence time of the GPS/BDS combination can be improved with 5, 15 and 30 min at the E, N and U components, respectively. For the Huawei P40, compared to GPS-only and BDS-only solutions, the convergence time of the GPS/BDS combination can be shortened, and the convergence time is about 5 min, 30 min and 5 min in the E, N, and U directions, respectively. The convergence time of multi-GNSS real-time static PPP is faster than that of a single system. In total, compared to GPS-only and BDS-only solutions, the positioning accuracy of dual frequency can be improved.

The RMS of static real-time PPP positioning accuracy using five combinations for Huawei Mate40, P40 and Novatel receivers in the E, N, and U components are listed in Table 5. As can be seen from the table, the real-time positioning accuracy of the Huawei P40 outperforms the Huawei Mate40 in terms of GPS-only and GPS/BDS combination. The positioning accuracy of the GPS/BDS/Galileo combination is the best among the five combinations, with RMS of 0.09, 0.27 and 0.12 m in the E, N and U components, respectively.

**Table 5.** RMS of real-time static PPP accuracy.

| System | Mate40 (m) | | | P40 (m) | | | DH04 (m) | | |
|---|---|---|---|---|---|---|---|---|---|
| | **East** | **North** | **Up** | **East** | **North** | **Up** | **East** | **North** | **Up** |
| G | 0.75 | 0.64 | 1.33 | 0.23 | 0.68 | 1.39 | 0.29 | 0.56 | 0.88 |
| C | 0.37 | 0.39 | 0.23 | 0.49 | 0.60 | 0.67 | 0.24 | 0.39 | 0.58 |
| GC | 0.13 | 0.15 | 0.21 | 0.28 | 0.28 | 0.30 | 0.24 | 0.36 | 0.55 |
| GCE | 0.08 | 0.13 | 0.20 | 0.09 | 0.27 | 0.12 | 0.23 | 0.33 | 0.54 |
| G (L1/L5) | 0.07 | 0.13 | 1.13 | 0.19 | 0.61 | 1.03 | 0.24 | 0.52 | 0.80 |

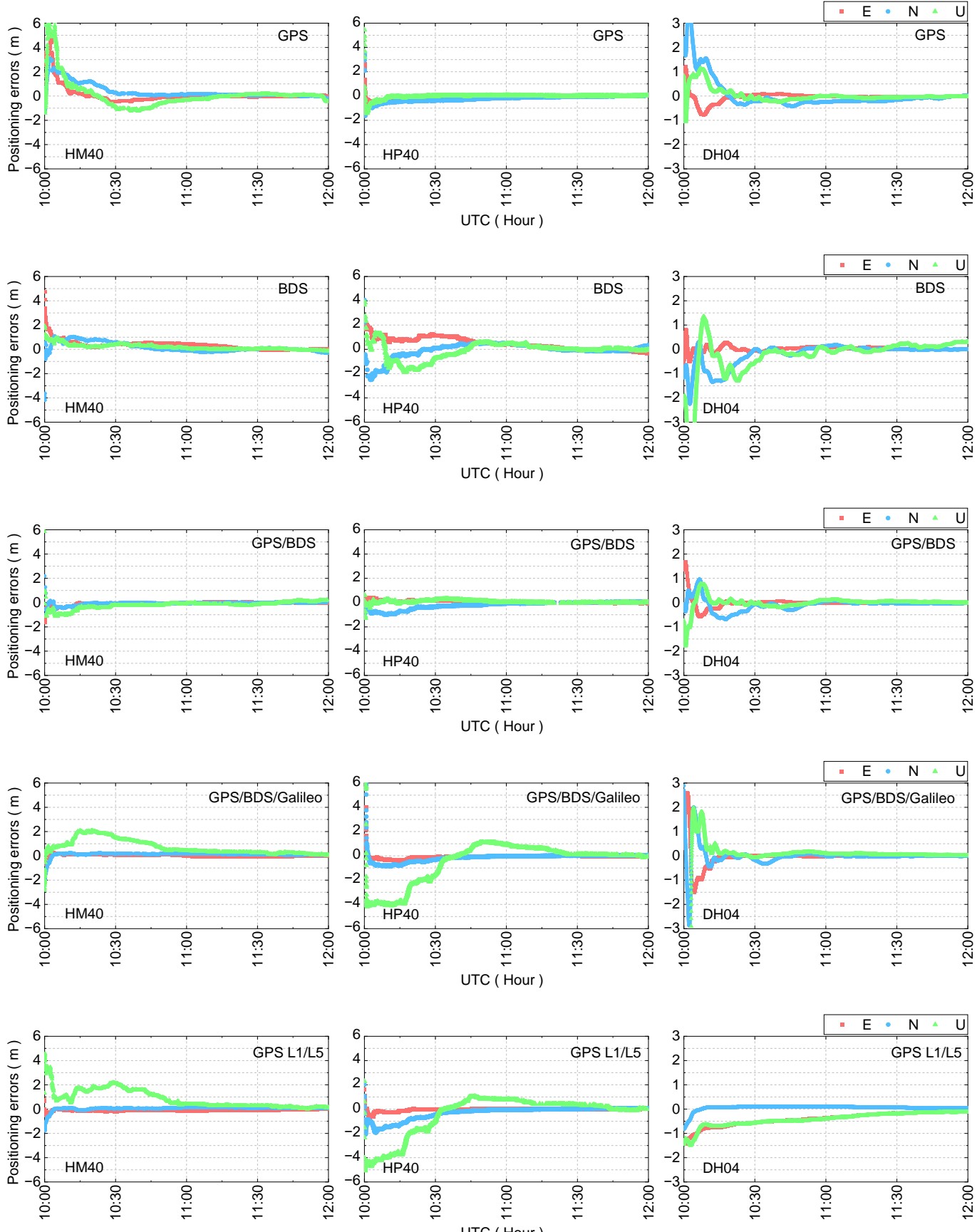

**Figure 9.** Real-time static PPP accuracy for Huawei Mate40, P40 and Novatel.

For the Huawei Mate40, compared to the GPS-only solution, the positioning accuracy of the GPS/BDS combination was improved from 0.75, 0.64 and 1.33 m to 0.13, 0.15 and 0.21 m in the E, N and U components, respectively, with an improvement of 83.42%, 77.12% and 84.16%, while the improvements are 66.40%, 62.18% and 9.48% compared to the BDS-only solution, respectively. Moreover, compared to GPS/BDS combination, the positioning accuracy of the GPS/BDS/Galileo combination was improved to 0.08, 0.13 and 0.20 m, with an improvement of 40%, 8.90% and 3.81% in three components. Compared to the single-frequency GPS-only solution, the positioning accuracy of 0.07, 0.13 and 1.13 m can be achieved for dual-frequency GPS L1/L5 combination in the E, N and U directions, respectively, with an improvement of 90.98%, 79.47% and 15.01%.

For the Huawei P40, the positioning accuracy of single-frequency GPS-only is 0.23, 0.68 and 1.39 m, compared to single-frequency GPS-only; the GPS/BDS combination can be improved by about 58.41% and 78.41% in the N and U components, while the improvement is 43.12%, 53.16 and 55.64% in the E, N and U components compared to BDS-only solution (0.49, 0.60, 0.67m), respectively. Compared to the GPS/BDS combination solution, the positioning accuracy of GPS/BDS/Galileo is 0.19, 0.61 and 1.03 m, with an improvement of 67.87%, 3.90% and 59.20% in the E, N and U components, respectively. For dual-frequency real-time static PPP, compared to the single-frequency GPS-only solution, the GPS L1/L5 combination can be improved by 19.83%, 10.18% and 25.63% in the E, N and U components, respectively. For the Novatel geodetic receiver, i.e., DH04, the decimeter-level positioning accuracy can be achieved for five combinations in east, north and up directions, respectively.

*5.2. Kinematic PPP Performance*

The time series of the kinematic real-time PPP positioning errors for the Huawei Mate40, P40 and Novatel receiver in the E, N and U components are illustrated in Figure 10. As can be seen from Figure 10, for the smartphone kinematic real-time PPP, the positioning results are impacted by large measurement noise, and the positioning errors are not smooth, which is worse than that of the static counterpart, while for the positioning errors of the DH04 receiver, it is smoother than that of the smartphone. The positioning errors of the Huawei Mate40 and P40 can achieve meter-level after convergence; it can also be found that the Huawei Mate40 is affected by system errors and shows the poorest positioning performance. In addition, compared to the Novatel receiver, the convergence time of the smartphone is long, and the positioning accuracy of the smartphone in the E and N component is much smaller than that in the U component. Compared to GPS-only, BDS-only and GPS/BDS combination solutions, the positioning performance of the GPS/BDS/Galileo combination can be improved.

The RMS of kinematic real-time PPP positioning accuracy of five combinations for the Huawei Mate40, P40 and Novatel receiver in three components are presented in Table 6. It can be clearly seen that the positioning accuracy of the Huawei P40 outperforms the Huawei Mate40. For Huawei Mate40 and P40, the positioning accuracy of the GPS/BDS combination solution is 3.47, 4.30, 8.57 m and 4.20, 3.49 and 8.07 m for E, N and U directions, which is better than that of GPS-only and BDS-only solutions. However, compared to the GPS/BDS combination solution, the improvements of GPS/BDS/Galileo solutions are not obvious, which may be because the observed Galileo satellite is small, and the contribution of the Galileo satellite is less. For dual-frequency, GPS L1/L5, the improvements in positioning accuracy are few compared to single-frequency GPS-only solutions, with an improvement of 2.49%, 1.38% and 3.26% in the east, north and up components, respectively. The positioning accuracy of the GPS/BDS/Galileo combination using the Huawei P40 smartphone is the best of the three components; its RMS is 3.88, 3.39 and 7.83 m in the three components, respectively. For the DH04 receiver, the positioning accuracy at a sub-centimeter level can be achieved for most solutions, in which the dual-frequency GPS L1/L5 is the best, and the positioning accuracy is 0.06, 0.11 and 0.16 m in E, N and U directions. Compared to real-time smartphone PPP, the positioning accuracy is significantly improved since it has low observation noise.

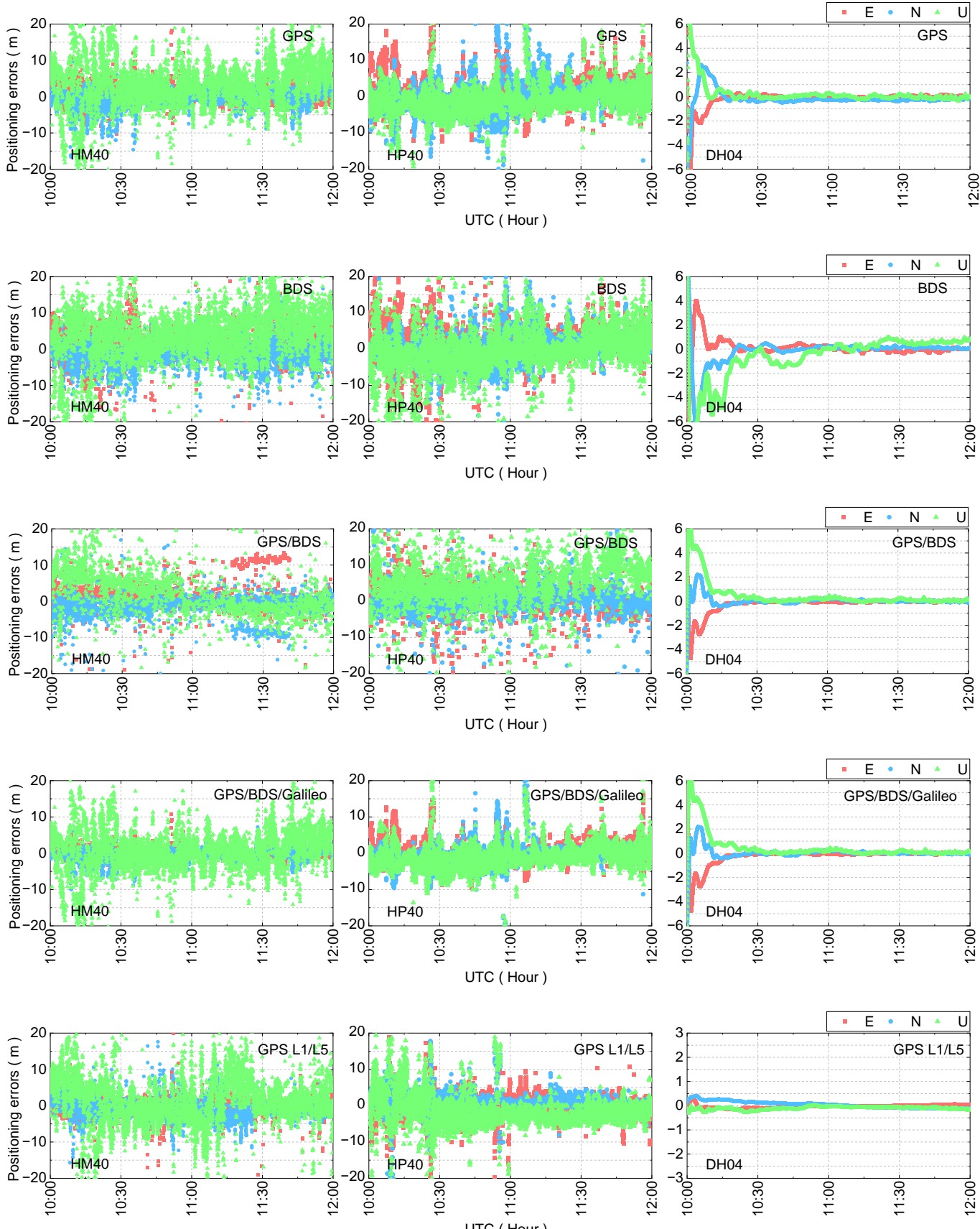

**Figure 10.** Real-time kinematic PPP accuracy for Huawei Mate40, P40 and Novatel.

**Table 6.** RMS of real-time kinematic PPP accuracy.

| System | Mate40 (m) | | | P40 (m) | | | DH04 (m) | | |
|---|---|---|---|---|---|---|---|---|---|
| | East | North | Up | East | North | Up | East | North | Up |
| G | 6.89 | 7.36 | 8.51 | 5.54 | 5.94 | 9.00 | 0.63 | 0.57 | 0.72 |
| C | 4.22 | 3.34 | 9.96 | 4.57 | 3.73 | 8.89 | 0.67 | 0.86 | 1.47 |
| GC | 3.47 | 4.30 | 8.57 | 4.20 | 3.49 | 8.07 | 0.58 | 0.40 | 0.89 |
| GCE | 3.45 | 4.41 | 8.60 | 3.88 | 3.39 | 7.83 | 0.47 | 0.53 | 0.84 |
| G (L1/L5) | 6.85 | 7.30 | 8.15 | 5.30 | 5.82 | 8.80 | 0.06 | 0.11 | 0.16 |

## 6. Conclusions

In this paper, the observations from Huawei Mate40, P40 and geodetic receivers are used, and real-time orbit and clock offset products are applied to conduct smartphone PPP. The following conclusions are drawn:

(1) The positive correlation between C/N0 and elevation is not found, the C/N0 of some observations is low at a higher elevation, and the degree of dispersion for smartphone C/N0 is larger than that of geodetic receivers. The C/N0 of GPS is the best, with an average value of 35 dB-Hz; the results of BDS and GLONASS are the second, with an average of 33 dB-Hz; the result of Galileo is the worst, with an average of 30 dB-Hz.

(2) The total number of satellites that can be observed by Huawei Mate40 and P40 is around 30. The number of tracked BDS satellites of the smartphone is larger than that of the GPS, while the GLONASS and Galileo system is relatively small. The PDOP of the Huawei P40 is slightly better than that of the Mate40, the smartphone PDOP of GPS shows the worst, while the average PDOP of BDS, GPS/BDS, GPS/BDS/GLONASS and GPS/BDS/GLONASS/Galileo combinations is less than 1.8.

(3) In terms of real-time orbit accuracy: the accuracy of most GPS and Galileo satellites is better than 0.05 m and 0.1 m in along, cross and radial directions, respectively. Different GLONASS satellites have smaller differences in along and cross directions, but the accuracy in the radial direction is larger. In terms of real-time clock offset accuracy: the clock offset accuracy of GPS and Galileo satellites is better than 0.15 and 0.25 ns, respectively. Compared with BDS-2, the clock offset accuracy of the BDS-3 satellite has been significantly improved, and is better than 0.28 ns, while the clock offset accuracy of the GLONASS satellite is better than 0.8 ns.

(4) In terms of real-time static PPP, the smartphone can achieve decimeter-level PPP accuracy after convergence. The GPS/BDS/Galileo combination of Huawei P40 shows the best PPP accuracy in three components, with RMS of 0.09, 0.27 and 0.12 m in the east, north and up components, respectively. Moreover, in terms of real-time kinematic PPP, there is a large difference between the PPP accuracy of smartphones and receivers, and kinematic, real-time PPP can achieve meter-level positioning accuracy. The GPS/BDS/Galileo combination of Huawei P40 presented the best PPP accuracy, with 3.88, 3.39 and 7.83 m in the E, N and U components, respectively.

In summary, there is a need for further development of real-time PPP for smartphones, but it is currently limited by the quality of the observation data. Hence, it is possible to achieve better real-time PPP accuracy by fusing positioning with multiple sensors to solve the drawbacks of single positioning in the future.

**Author Contributions:** M.L. and G.H. conceived and designed these experiments and wrote this paper; M.L., G.H., L.W. and W.X. performed the experiments, analyzed the data, drew pictures, and wrote this paper. All authors have read and agreed to the published version of the manuscript.

**Funding:** This work was supported by the Programs of the National Natural Science Foundation of China (42127802), the Key R&D Program of Shaanxi Province (2022ZDLSF07-12), the Special Fund for Basic Scientific Research of Central Colleges (Grant No. CHD300102269305, CHD300102268305, Chang'an University).

**Data Availability Statement:** The observation data and precise products used in the research are available on the FTP of Wuhan University (ntrip.gnsslab.cn; 112.65.161.226).

**Acknowledgments:** The IGS and GFZ are greatly acknowledged for providing the multi-GNSS tracking data, SINEX coordinates, and satellite orbit and clock offset products. We would also like to thank Zhongyang Zhao, an employee of Xi'an Honor Device Co., Ltd. for his suggestions on this paper.

**Conflicts of Interest:** The authors declare no conflict of interest.

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
