# Peer review of "BDS/GPS/Galileo Precise Point Positioning Performance Analysis of Android Smartphones Based on Real-Time Stream Data"

_remotesensing, doi:10.3390/rs15122983_

Round 1

Reviewer 1 Report

Review

The study evaluated the performances of smartphone precise point positioning using real-time products. These efforts have important implications in terms of smartphone positioning. The presentation of the paper is clear. Besides, the method and results have reference significance for subsequent research. The followings are some suggestions to further improve the quality of the manuscript.

1.       In Figure 6, the vertical ordinate name should be “PDOP”, please modify.

2.       Please explain the reason to choose the real-time products of GFZ. Do the authors try the real-time products from CNES, since CNES is well-known for their real-time products.

3.       How is the phone placed? Up or downward? Please clarify in the manuscript.

4.       In Table 2, the time is suggested to be Release time.

5.       In the right bottom of figures 3 and 4, the legend is only red and green lines. However, there is no legend for blue and brown lines, please clarify.

  English needs further check and minor revision

Reviewer 2 Report

This paper studied the performance of smartphone precise point positioning (PPP) using real-time stream data. The observation quality of smartphones and the performance of the real-time product were comprehensively evaluated. Meanwhile, the static and kinematic PPP performance of single and dual-frequency multi-system were evaluated in detail. This paper can provide a reference for smartphones applied for GNSS positioning. But there are some problems, which must be solved before it is considered for publication.

(1)  In Table 2, the Galileo system for Android smartphones can receive the dual-frequency E1 and E5a currently, please revised.

(2)  In the GPS dual-frequency PPP, the L1/L5 signal is applied, while the clock offset is estimated from L1/L2, how to deal with the inter-frequency clock bias? Ignored or corrected?

(3)  What kind of estimator was used in this manuscript?

(4)  In Table 3, it is clock offset STD, not clock offset, please revised.

(5)  There are spaces between numbers and units, or there are no spaces between them. Please standardize the format.

(6)  The reference format is not uniform, please revised.

This paper studied the performance of smartphone precise point positioning (PPP) using real-time stream data. The observation quality of smartphones and the performance of the real-time product were comprehensively evaluated. Meanwhile, the static and kinematic PPP performance of single and dual-frequency multi-system were evaluated in detail. This paper can provide a reference for smartphones applied for GNSS positioning. But there are some problems, which must be solved before it is considered for publication.

(1)  In Table 2, the Galileo system for Android smartphones can receive the dual-frequency E1 and E5a currently, please revised.

(2)  In the GPS dual-frequency PPP, the L1/L5 signal is applied, while the clock offset is estimated from L1/L2, how to deal with the inter-frequency clock bias? Ignored or corrected?

(3)  What kind of estimator was used in this manuscript?

(4)  In Table 3, it is clock offset STD, not clock offset, please revised.

(5)  There are spaces between numbers and units, or there are no spaces between them. Please standardize the format.

(6)  The reference format is not uniform, please revised.

Reviewer 3 Report

The research looks like bachelor degree level investigations and experiments. No innovations are suggested. However, a lot of smartphones users could be interested in positioning possibilities using their phones, so paper could be printed.

Some similarities could be found, look into attached report. I understand, that mostly they are terms, however nowadays the checking softwares are very strict.
